# Bacterial Composition and Interactions in Raw Milk and Teat Skin of Dairy Cows

Hui Yan [1,2,†], Wen Du [1,†], Shoukun Ji [1,2], Chunyan Guo [1,3], Yujing Zhang [4], Yajing Wang [1], Zhijun Cao [1] and Shengli Li [1,*]

1   State Key Laboratory of Animal Nutrition, Beijing Engineering Technology Research Center of Raw Milk Quality and Safety Control, College of Animal Science and Technology, China Agricultural University, Beijing 100193, China; yanhuihui@126.com (H.Y.); duwen1022@126.com (W.D.); jishoukun@163.com (S.J.); jzzygcy@126.com (C.G.); wangyajing2009@gmail.com (Y.W.); caozhijun@cau.edu.cn (Z.C.)
2   College of Animal Science and Technology, Hebei Agricultural University, Baoding 071001, China
3   Jinzhong Vocational and Technical College, Jinzhong 030600, China
4   Faculty of Veterinary and Agricultural Sciences, The University of Melbourne, Parkville, VIC 3010, Australia; zhang15@student.unimelb.edu
*   Correspondence: lisheng0677@163.com
†   These authors contributed equally to this work.

**Abstract:** The microbiota in raw milk plays an important role in the health of dairy cows and the safety of dairy products, which might be influenced by that in teat skin. However, the microbiota composition in raw milk and teat skin, as well as the bacterial interaction between the two adjacent spatial locations, remains elusive. Here, we investigated the composition, diversity, and co-occurrence network of the bacterial communities in raw milk and on teat skin, as well as the shift of bacterial communities during the teat bath using 469 samples from 156 individual cows. We observed that raw milk and teat skin harbored significantly different bacterial communities according to an assessment of the genera numbers ($p < 0.05$) and PCoA analysis (ANOSIM $p < 0.05$). The microbiota in raw milk was dominated by Proteobacteria (58.5% in relative abundance) at the phylum level and by *Pseudomonas* (51.2%) at the genus level, while that in teat skin was dominated by Firmicutes (46.9%) at the phylum level and by *Pseudomonas* (11.0%) at the genus level. We observed a massive difference between the bacterial subnetworks in raw milk and teat, and the bacterial abundance in these two adjacent spatial locations was positively correlated ($p < 0.05$). Using Bayesian algorithms, we identified that 92.1% of bacteria in raw milk were transferred from teat skin, while 63.6% of bacteria on teat skin were transferred from raw milk. Moreover, microbiota composition in teat skin could be affected by the teat bath with iodine disinfectant, which tended to be more similar to that in raw milk after the teat bath ($p < 0.05$), while the abundance of the dominant genus *Pseudomonas* significantly increased ($p < 0.05$). These findings expand our knowledge on the microbiota composition in teat skin and raw milk, as well as the interaction between these two adjacent spatial locations.

**Keywords:** bacteria composition; bacteria interaction; raw milk; teat skin; *Pseudomonas*

## 1. Introduction

Raw milk harbors a robust microorganism ecosystem, which has been proven to be not only related to the udder health of cows [1], but may also affect food safety and fermentation, as bacteria in milk might change the quality of milk products [2], and milk is susceptible to microbe-induced spoilage even after pasteurization or ultrahigh-temperature (UHT) processes [3]. Therefore, there is increasing concern, as and the microbiota profiles in raw milk of dairy cows will have a profound implication for dairy husbandry and the food industry.

The microbiota diversity in raw milk has been identified in previous studies [4,5]; however, where the microbiota in raw milk comes from is still a controversial issue. A

previous study indicated that the gut microbiota might transferred to milk through the entero-mammary pathway [6], but another study demonstrated that teat skin might also be a potential source of the milk microbiota [7], while the microbiota composition in raw milk might also be affected by the environment where the cows lived [8]. Teat skin, adjacent to the mammary duct, has demonstrated a key role as a reservoir of microbial diversity for raw milk, but previous studies on the proportion of bacteria in raw milk sourced from the teat skin were not consistent [9]. Understanding the interaction of bacteria between raw milk and teat skin might be helpful for controlling the microbiota in raw milk.

A teat bath using disinfection products is an essential step for the milking procedure, which has been demonstrated to efficiently reduced the bacterial load on teat skin and in raw milk [10]. Among these disinfection products, iodine-based teat disinfectant is most used for the prevention of intramammary infections in dairy cows, but different bacterial taxa are not equally sensitive to the iodine-based teat disinfectant [11,12]; thus, the influence of iodine-based teat disinfectant on the microbiota of teat skin is not well understood.

Here, 469 raw milk and teat skin samples were collected from 156 dairy cows, and the microbiota was investigated with high-throughput sequencing of 16S ribosomal RNA (rRNA) genes. We aimed to describe the bacterial profiles and bacterial interactions in raw milk and on teat skin, as well as monitor the dynamic changes of the bacterial taxa on the teat skin during the teat bath process.

## 2. Materials and Methods

### 2.1. Sample Collection

Milk and teat skin swab samples were collected from healthy Holstein cows ($n = 156$) during their early lactation (45 ± 15 days in milk) at Fengning Farm (Hebei, China) and Hengshui Farm (Hebei, China) in October 2016. The first three squirts were discarded, and then milk samples were collected from 156 individual cows before the pre-milking teat bath. The pre-milking teat bath was operated with iodine-based teat disinfectant (1% *w/v*), and the teat was dried with a paper towel. After milking, a post-milking teat bath was carried out with iodine-based teat disinfectant (2% *w/v*). Teat skin was swabbed with sterilized cotton swabs moistened with physiological saline before the pre-milking teat bath (Skin A, $n = 111$), after the pre-milking teat bath (Skin B, $n = 115$), and after the post-milking teat bath (Skin C, $n = 87$). Milk and teat skin swab samples were immediately transferred into separate sterilized sampling tubes and snap-frozen in liquid nitrogen, before storing at −80 °C until analysis.

### 2.2. DNA Extraction and High-Throughput Sequencing of 16S rRNA Amplicon

The genomic DNA of the milk and teat skin microbial samples were extracted using the CTAB-based methods for bacteria. The quality of extracted DNA was detected by 1% agarose gels, and concentrations were detected by Nanodrop ND-2000 and diluted to 1 ng/μL to make the quality and cycles of amplification uniform. The V3 variable region of 16S rRNA genes was used for further analysis.

### 2.3. Sequencing Data Processing

The Cutadapt software (version 1.15) [13] was used to trim off adapters, barcodes, and primers. Trimmomatic-0.36 [14] was used to trim off low-quality ends of sequences using a 50 nt sliding window with an average quality value under 25 to remove read lengths shorter than 90 bp. Clean reads were assembled using FLASH-1.2.11 by joining the overlap between paired-end reads [15]. Chimera detection was performed using usearch6.1 [16], and non-chimera sequences were extracted. The sequence numbers of each sample were normalized to 40,000 reads according to the random selection principle. The V3 region was uniformly obtained using Perl script for further analysis. Then, all sequences from the 469 samples were combined into a single file and were pushed into the QIIME pipeline [17]. Sequences were clustered into OTUs on the basis of 97% identity using the uclust algorithm [16]. Representative sequences were picked out and assigned to taxonomy

against the GreenGenes database [18]. Alpha diversity and beta diversity (Bray-Curtis distance) were calculated via the QIIME pipeline [17].

### 2.4. Core Microbiota and Network Analysis

Core bacteria in raw milk and teat skin before the pre-milking teat bath were identified as the bacteria present in more than 80% of samples in each group, and predominant bacteria were defined as bacteria with a relative abundance higher than 2%, from which we were able to show the main milk and teat skin bacterial community. We constructed a meta-network of the bacterial community in milk and on teat skin before the pre-milking teat bath on the basis of their core bacterial communities at the genus level using the igraph R packages (version 1.0.1). The network was inferred from the Spearman correlation matrix calculated by R software (version 3.3.0). The nodes in this network represent genera, while the edges that connect these nodes represent correlations between genera. According to the false discovery rate (FDR)-adjusted *p*-values of the correlation with a cutoff threshold value at 0.001 [19], the subnetworks of the bacterial community in milk and on teat skin were constructed independently. Then, the interactions of the same genera in paired collected milk and teat skin samples were calculated. A meta-network was constructed from the subnetworks and their interaction edges. The topological features of the meta-network and subnetworks were calculated using igraph R packages.

### 2.5. Bacterial Transfer Analysis

SourceTracker (version 1.0.1) was developed to track microbial sources from surrounding environments using a Bayesian approach [20]. We performed SourceTracker analysis to provide quantitative insights into microbial ecology in milk and on teat skin before the pre-milking teat bath with alternate inputs as source samples and sink samples. This analysis provided a further understanding of the bacterial transfer between these two adjacent habitats.

### 2.6. Statistical Methods

Principal coordinates analysis (PCoA) was performed on the basis of Bray-Curtis distance, and analysis of similarities (ANOSIM) was performed to identify the bacterial community differences between groups using the Vegan packages (version 2.4.1) in R software (version 3.3.0). Venn plots and hypergeometric tests were used to present the distribution difference of the core bacterial community with the VennDiagram package (1.6.17). The node distribution with a binomial distribution and power-law distribution were analyzed using R (version 3.3.0). The binomial distribution was formulated as $y = ax + bx^2 + c$, and the power-law distribution was formulated as $y = ax^b + c$, where x is the node number, and y is the proportion of nodes. The better fitting formula with higher correlation coefficients was chosen to describe the network pattern. The Spearman correlation test was used to examine the correlation between relative abundance and node centrality, as well as the relative abundance and proportion of bacterial transfer using R (version 3.3.0). The groups were compared using the Mann-Whitney U test or the Kruskal-Wallis test. A *p*-value of <0.05 was considered significant; however, with multiple comparisons in bacterial analysis, the FDR adjusted *p*-value was used.

## 3. Results

### 3.1. The Bacterial Communities in Milk and on Teat Skin

We obtained 18,685,992 sequences (39,842 ± 2115) of the V3 region from a total of 469 samples (313 teat skin samples and 156 raw milk samples), and 101 paired collected milk and skin samples before the pre-milking teat bath were included without the influence of the teat bath. With these 101 paired collected milk and skin samples, we observed high bacterial richness in both milk and skin samples with milk samples harboring 332.3 ± 80.6 bacterial genera and skin samples harboring 385.5 ± 72.2 bacterial genera; the number of bacterial genera on the skin was significantly higher than that in milk samples ($p < 0.05$; Figure 1a). The bacterial communities in milk and on skin were clustered separately in the PCoA as

a function of the Bray-Curtis distance (ANOSIM, $p < 0.05$; Figure 1b,c). Moreover, the bacterial community on the skin also showed a higher within-group dissimilarity than that in milk ($p < 0.05$; Figure 1c), indicating that the bacterial community on the skin might be more variable than that in milk.

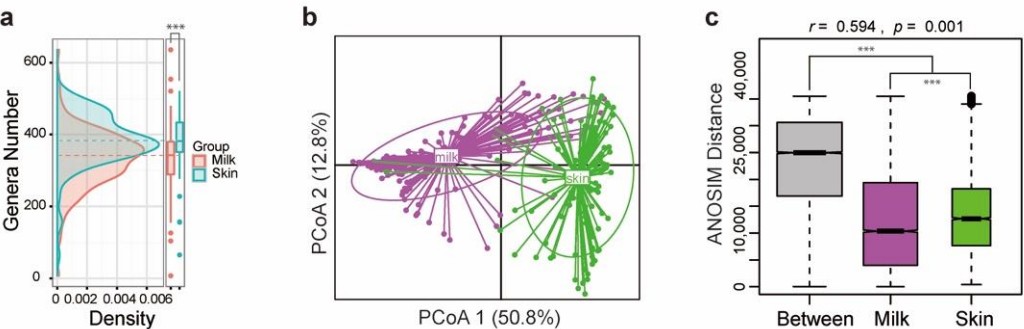

**Figure 1.** Bacteria composition in raw milk and on teat skin before pre-milking teat bath. (**a**) The numbers of microbial genera. (**b**) The PCoA of the bacterial community based on Bray-Curtis distance. (**c**) ANOSIM of the bacterial community; *** $p < 0.01$.

The bacteria in both milk and skin swab samples before the pre-milking teat bath were dominated (>2% in abundance) by Proteobacteria, Firmicutes, Actinobacteria, and Bacteroidetes at the phylum level (Figure 2a,c). Notably, Proteobacteria was the most dominant phylum in milk (58.5%), while Firmicutes was the most dominant phylum on the skin (46.9%). The abundance of Actinobacteria and Bacteroidetes on the skin was also higher than that in milk (FDR < 0.05). We observed four and 11 dominant genera in milk and on skin at the genus level, respectively (Figure 2b,d), and the most predominant genus in both habitats was *Pseudomonas*; however, the abundance in milk (51.2%) was much higher than that on the skin (11.0%; FDR < 0.05).

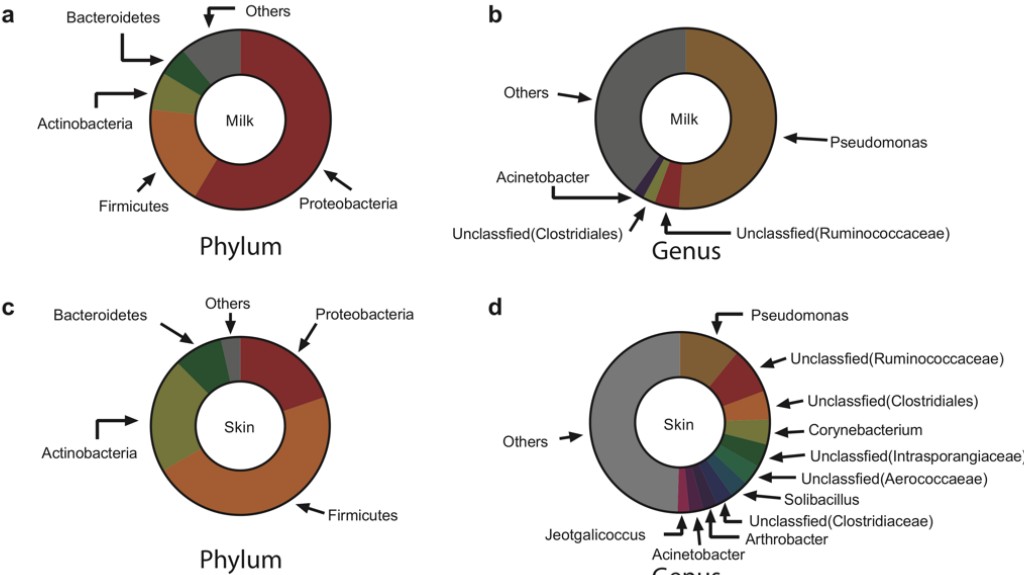

**Figure 2.** Predominant bacteria in raw milk and on teat skin before pre-milking teat bath. (**a**) Dominant bacteria at the phylum level in raw milk. (**b**) Dominant bacteria at the genus level in raw milk. (**c**) Dominant bacteria at the phylum level on teat skin. (**d**) Dominant bacteria at the genus level on teat skin.

### 3.2. The Core Microbiota in Raw Milk and on Teat Skin

We studied the core microbiome according to an 80% presence in milk and on teat skin before the pre-milking teat bath (Figure S1a). A total of 210 core genera were detected in

the meta-dataset of milk and skin samples. Among these core bacterial genera, 152 genera were observed in milk samples, which accounted for 72.4% of the total detected core genera (210 genera) and 11.6% of total observed genera (1316 genera); 205 core bacterial genera were observed in skin samples, which accounted for 97.6% of the total detected core genera (210 genera) and 18.4% of the total observed genera (1316 genera). These core bacterial genera, either from milk or skin, mainly belonged to Actinobacteria, Bacteroidetes, Firmicutes, and Proteobacteria at the phylum level. Although skin harbored more core bacterial genera than milk, the distribution of core bacterial genera in milk and on skin showed no significant difference according to the hypergeometric test ($p > 0.05$), as milk and skin shared 70.0% (147 out of 210 genera) common core genera (Figure S1b).

### 3.3. Co-Occurrence Network of the Bacterial Community in Milk and on Teat Skin

We constructed the co-occurrence network using the bacterial genus dataset from paired collected samples to reveal the natural communication relationship between raw milk and the teat skin. In total, 7473 associations (edges) among 421 microbial genera (nodes) were constructed, and 58.6% and 40.4% of edges were identified as within-group connections in the milk or skin subnetwork, respectively; the remaining 1.0% of edges were identified as inter-group connections between the milk and skin microbiota. In the milk sub-network, 96.6% of edges were identified as positive connections, and only 3.4% of edges were identified as negative connections; a relatively lower proportion of positive connections (86.3%) and a higher proportion of negative connections (13.7%) were detected in the skin subnetwork, while all 72 inter-group connections between raw milk and teat skin were detected as positive associations. The number of nodes in the subnetwork in milk (210 nodes) and on skin (211 nodes) was similar, and five hub nodes with most edges were identified, which belonged to *Pseudomonas*, *Arthrobacter*, and three unclassified genera belonged to Aerococcaceae, Mogibacteriaceae, and Micrococcaceae. These hub nodes were all located in the subnetwork of milk, reflecting that bacteria in milk were more extensively connected than those on the skin. (Figure 3a).

To illustrate the network patterns, we fitted the node distribution with a binomial distribution and power-law distribution. The degree distribution of the microbial meta-network and subnetwork on the skin followed a power-law distribution (Figure 3b); in comparison, the milk subnetwork was characterized as a random structure with the degree distributed roughly following a binomial distribution (Figure 3b). We then estimated the topological features of subnetworks by assessing the network centrality with the degree, betweenness, and closeness indices. Although similar node numbers were included in the subnetworks of milk (210) and skin (211), the degree centrality and the closeness centrality of the subnetwork in milk were significantly higher than that on the skin, while the betweenness centrality was similar in the two subnetworks (Figure 3c).

We further assessed whether bacterial abundance influenced the centrality in milk and on skin, respectively. Most of the centrality indices (betweenness and closeness centrality in milk, degree and betweenness centrality on the skin) were positively correlated with relative bacterial abundance ($p < 0.01$, Spearman correlation test). Degree centrality in milk and closeness centrality on the skin were not significantly correlated with relative bacterial abundance ($p > 0.05$, Spearman correlation test) (Figure 3d–f).

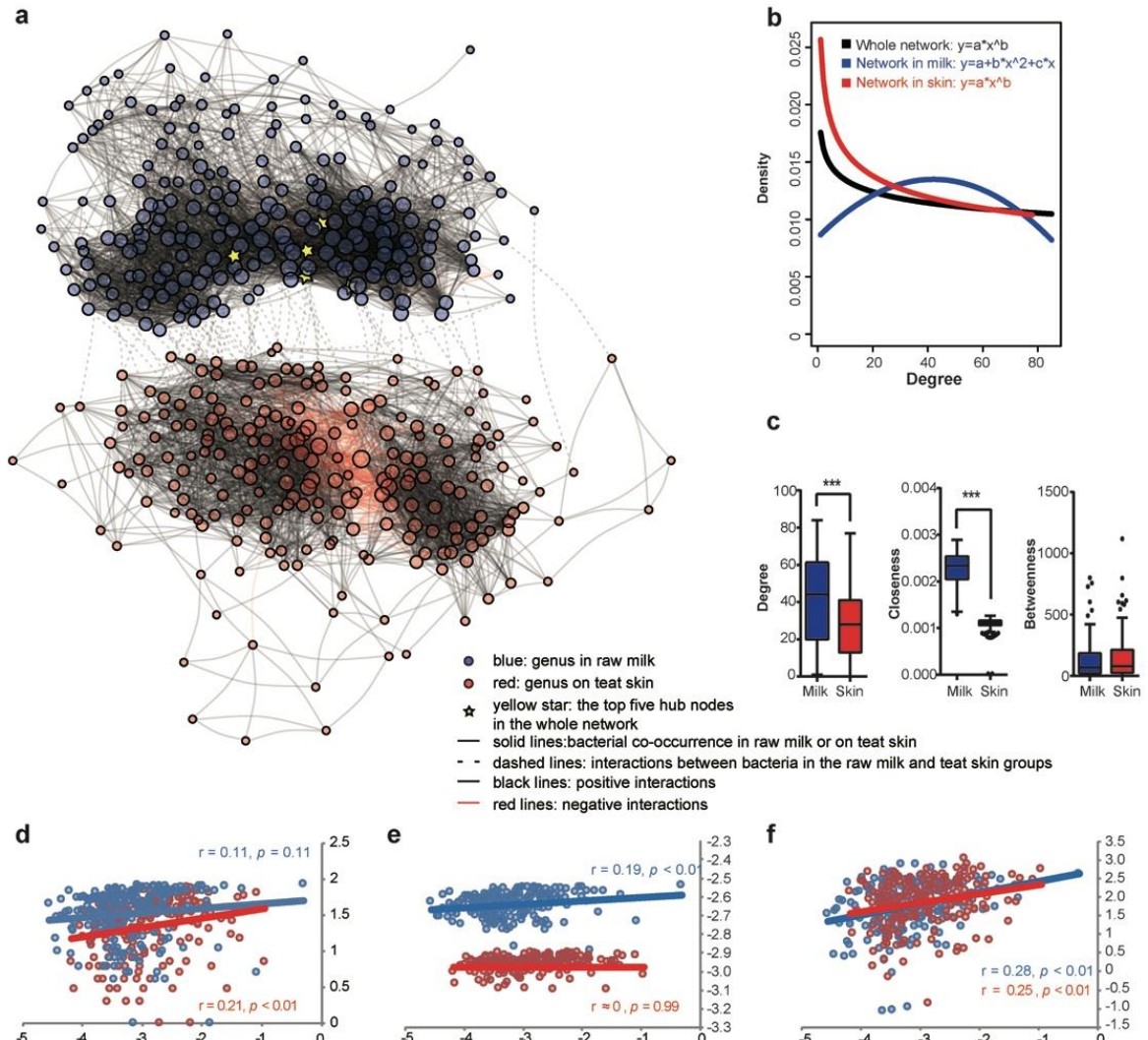

**Figure 3.** Co-occurrence network interactions of microbiota in raw milk and on teat skin before pre-milking teat bath. (**a**) Co-occurrence network of the meta-dataset of bacterial taxa at the genus level. The nodes represent the bacterial taxa at the genus level; the diameter of the node is consistent with its degree. (**b**) The distribution of nodes in the whole co-occurrence network, raw milk subnetwork, and teat skin subnetwork based on the degree centrality. (**c**) Degree centrality, closeness centrality, and betweenness centrality of subnetworks of raw milk and teat skin. (**d–f**) The correlation between bacterial abundance and degree centrality (**d**), closeness centrality (**e**), and betweenness centrality (**f**) at the genus level; *** $p < 0.01$.

### 3.4. Bacterial Transfer between the Raw Milk and Teat Skin

To further assess the interactions of the bacterial community in milk and on the skin, we calculated the transfer proportion of bacteria between raw milk and teat skin before the pre-milking teat bath using an iterative Bayesian approach with the default parameters of the SourceTracker software (version 1.0.1) [20]. We identified that $92.1\% \pm 18.6\%$ of the bacteria in milk might be potentially transferred from the skin, while a much lower proportion of the bacteria on the skin ($63.6\% \pm 23.0\%$) might be originated from milk (Figure 4a). The high proportions of transferred bacteria between milk and skin were mainly contributed by Actinobacteria, Bacteroidetes, Firmicutes, and Proteobacteria at the phylum level, which were also present as dominant bacteria in milk and on skin (Figure S2). A Spearman correlation analysis was further performed to calculate the interactions between the transfer contribution and relative abundance of bacteria in milk and on the skin. Significant positive correlations ($p < 0.05$) were observed between the relative abundance

of bacteria communities and transfer contributions, which means that highly abundant bacteria were more active in transferring between the two habitats (Figure 4b).

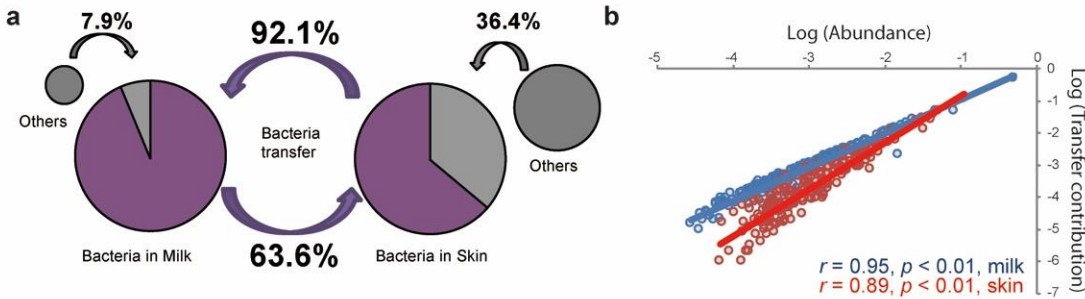

**Figure 4.** Bacteria transfer between raw milk and teat skin before the pre-milking teat bath. (**a**) The proportion of bacteria transferred between raw milk and teat skin. The pie chart on the left represents bacteria in milk, while that on the right represents bacteria on teat skin. The purple section is the proportion of bacteria transferred from either teat skin or raw milk. (**b**) The correlation between bacterial abundance and transfer proportion at the genus level; the red circles and blue circles represent bacteria in teat skin and raw milk, respectively.

### 3.5. Bacterial Taxa Shift on Teat Skin during Milking

In the standard operating procedure for milking, a medicated teat bath is performed before and after milking to reduce the bacterial load on teat skin and prevent mastitis. Here, we monitored the dynamic change of the bacterial taxa on the teat skin of cows before (Skin A) and after (Skin B) the pre-milking teat bath, as well as after the post-milking teat bath (Skin C). Firstly, we observed a similar bacterial genera number distributed on teat skin at the three monitored timepoints (Figure 5a), while the mean bacterial abundance on Skin C was significantly higher than that on Skin A and Skin B ($p < 0.05$, Figure 5b). We then observed that the bacterial community in each group could be distinguished on the basis of PCoA using Bray-Curtis distance (ANOSIM $p < 0.05$, Figures 5c and S3), with the ANOSIM distance value of the Skin A group being the highest, while that of Skin C was the lowest (Figure 5d). These results demonstrate that the bacterial community on teat skin tended to become more consistent with that in milk during the milking procedure.

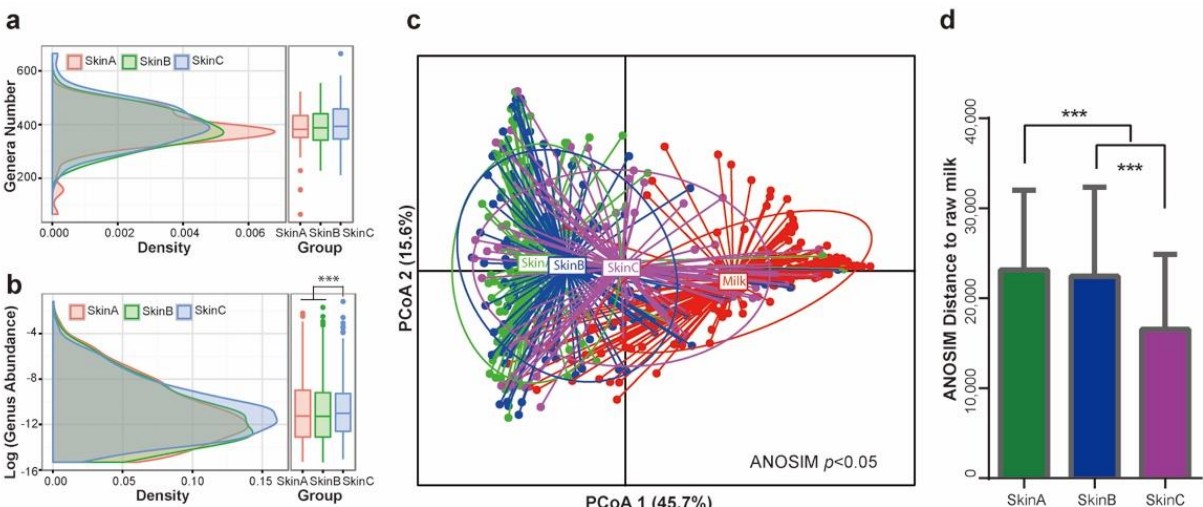

**Figure 5.** The microbiota changes on teat skin during the milking procedure. (**a**) The distribution of genus numbers. (**b**) The distribution of genus abundance. (**c**) PCoA of microbiota compared with that in raw milk based on Bray-Curtis distance. (**d**) The ANOSIM of the bacterial community. SkinA, teat skin from before the pre-milking medicated teat bath; SkinB, teat skin from after the pre-milking medicated teat bath; SkinC, teat skin from after the post-milking medicated teat bath. *** $p < 0.01$.



We further monitored the dynamic shift of core bacterial taxa on the skin at the genus level during the milking procedure. Most bacteria were not significantly changed after the pre-milking teat bath except *Pseudomonas*, belonging to Proteobacteria at the phylum level, which increased in abundance after the pre-milking teat bath (Figure 6a, FDR < 0.05). A total of 34 genera were significantly changed after the post-milking teat bath (Figure 6b, FDR < 0.05), which accounted for 16.6% of the number of core genera. Among these, the abundance of 29 (85.3%) genera was decreased, while that of five (14.7%) genera was increased. The decreased bacterial taxa mainly belonged to Actinobacteria, Bacteroidetes, and Firmicutes at the phylum level, while the enriched bacterial taxa all belonged to Proteobacteria at the phylum level.

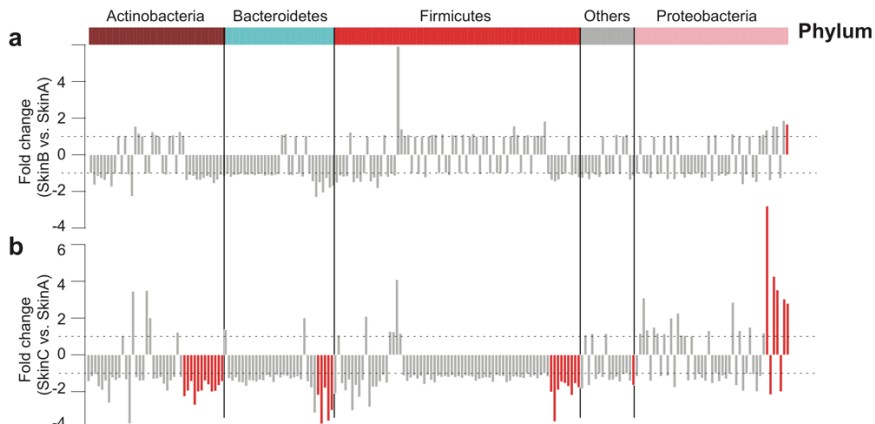

**Figure 6.** Change of bacterial abundance on teat skin during the milking procedure at the genus level. (**a**) Fold change of bacterial abundance in the after pre-milking medicated teat bath group (Skin B) compared with the before pre-milking medicated teat bath group (Skin A). (**b**) Fold change of bacterial abundance in the after post-milking medicated teat bath group (Skin C) compared with the before pre-milking medicated teat bath group (Skin A). The dashed line represents a onefold change, while the red bars represent a significant change (*p* < 0.05).

## 4. Discussion

A number of researchers have identified the diversity of bacteria in raw milk [5], in pasteurized milk [21], in fermented cheese [22], and on the teat skin [23] of dairy cows, highlighting the importance of the microbiota in raw milk and on teat skin. Understanding the microbiota in milk and on teat skin is critical for dairy husbandry and food safety [23–25]. In this research, using 469 samples from the milk and teat skin of dairy cows, we delineated the bacterial distribution, explored the bacterial interaction in these two adjacent habitats, and illustrated the shift of the bacterial community on teat skin during the teat bath procedure.

Bacteria in raw milk and on teat skin were richly observed by the higher resolution of high-throughput sequencing of 16S rRNA genes [3,8,26], which was far more diverse than observations using traditional cultures [27] or DGGE methods [28]. We observed Firmicutes, Proteobacteria, Actinobacteria, and Bacteroidetes as the dominant phyla in raw milk and on teat skin samples, but their abundance differed in the two adjacent locations. Even though there is close spatial proximity of milk and teat skin, different bacterial compositions in these adjacent habitats were identified; the bacterial ecology in milk was disproportionately dominated by a few taxa, while, on the skin, there was a more even distribution. *Pseudomonas* was observed to be the predominant genus both in raw milk and on teat skin in this study, which is consistent with previous observations [7,29]. However, some other researchers also observed different dominant genera in milk and teat skin, such as *Lactococcus* [21], *Lactobacillus* [30], *Propionibacterium* [29], or *Streptococcus* [26] in raw milk, and *Staphylococcus* [7,23] or *Corynebacterium* [31] on teat skin, which may be affected by diet and breed, as reported previously [7,23,28,30–33]. Core bacteria represented the taxa that were prevalent in a specific ecosystem [34], and we observed teat skin was expected

to harbor more core genera, although the two habitats shared 70.0% of core genera. This result revealed that bacteria in these two adjacent habitats are distinguishable communities, but they might be exchanged, as the bacterial exchange may induce a more similar bacterial community in distinct habitats [35].

Bacterial co-occurrence networks can reflect the ecological pattern of microbiota [36], and two kinds of co-occurrence networks, i.e., random structure or scale-free network structure, are mostly observed [37]. The scale-free network structure features most nodes having few connections and only a few hub nodes having multifarious connections, which is believed to have strong robustness and resilience abilities [38]. We observed a scale-free network ecosystem in teat skin microbiota but a random structure in raw milk; meanwhile, the degree and closeness indices of microbiota in milk were higher than those on teat skin. Previous studies demonstrated that nutrients [39] and the environment [40] are two main factors influencing the bacterial community, as the nutrition level of teat skin might be rhythmically changed during a milking process; thus, the teat skin was exposed to a much more open and complex external environment [41], which increased the niche capability, as well as the dispersal capability for xenomicrobiota colonizing, on teat skin [42]; thus, resident bacteria on teat skin experienced a regularly changing environment and challenges from new invaders with a more rigorous natural selection than that in teat cistern [43], which could induce a more robust and complex bacterial community on teat skin than that in raw milk.

The majority of interactions in the co-occurrence network were positive for both raw milk (96.6%) and teat skin (86.3%) microbiota, and the bacteria between teat skin and raw milk groups all positively interacted, indicating that most genera showed cooperative relationships rather than competitive relationships [44]. From these findings, we could infer that bacteria on teat skin and in raw milk were involved in coupled and positive feedbacks [45,46]. Another inference from this finding was that the bacteria in these two adjacent habitats of teat skin and teat cistern might be exchanged, because, under an unobstructed exchange system, bacterial change in one habitat may induce a coupled change in another adjacent habitat [47], which was also supported by the highly shared core bacteria in raw milk and on teat skin in this study.

Understanding the movement of microbes from the environment into the raw milk is important for controlling the transfer of bacteria into the food chain. A previous study with samples from three cows and their surrounding environment demonstrated that teat surface was the most significant source of contamination in raw milk, with herd feces being the next most prevalent source of contamination [8]. Vacheyrou et al. (2011) also speculated that the bacteria on the teat surface were the main sources of bacteria in raw milk for dairy cows by cultivation method [30]. These previous studies demonstrated the teat skin as the main source of bacteria in raw milk; however, due to the limited samples and huge individual variations, knowledge of the bacterial transfer between raw milk and teat skin was not quantitatively illustrated. We used 101 paired collected raw milk and teat skin swab samples before the pre-milking teat bath; we observed that a majority of bacteria (92.1% ± 18.6%) in raw milk were transferred from teat skin, and about 63.6% of bacteria on teat skin might be transferred from raw milk. Although we did not sample the bacterial community in the surrounding environment, the high bidirectional transfer proportion of bacteria between milk and teat skin might also indicate the positive correlation between bacteria in teat skin and the teat cistern, and targeting the bacteria in teat skin might be an efficient method in controlling the transfer of bacteria into raw milk.

A teat bath with iodine disinfectant before and after milking is a fundamental practice in the milking process to reduce the bacterial loads in raw milk and on teat skin to ensure the high quality of raw milk [48,49]. The microbial community on teat skin significantly changed after the pre-milking teat bath or post-milking teat bath, and the bacterial community on teat skin tended to be more similar to that in milk after the teat bath. The sensitivity of different bacterial taxa to the teat bath also differed, thus inducing a bias to the bacterial community on teat skin. The teat bath was more effective in reducing the

relative abundance of Actinobacteria, Bacteroidetes, and Firmicutes, but it was less effective at reducing the bacteria abundance belong to Proteobacteria, especially the overwhelming dominant genus *Pseudomonas*. *Pseudomonas* has also been stated to be associated with the pre-milking disinfectant [50] and was identified as a psychrotolerant bacteria with high metabolic activity [51], which was enriched during refrigeration [28,52]. In addition, the heat-resistant peptidase with a high proteolytic and lipolytic activity secreted by *Pseudomonas* also remains active after UHT processes, thus resulting in destabilization of milk [51,53,54]. Therefore, even though the teat bath with iodine-based disinfectant is efficient in reducing most bacteria taxa, a better disinfectant is still needed to restrict *Pseudomonas* in the future.

## 5. Conclusions

In conclusion, our work provided new insight into the bacterial communities and interactions between raw milk and teat skin. These findings highlighted the comprehensive communication between the bacterial communities in raw milk and on teat skin, suggesting that the bacterial community on teat skin might be a target for manipulating the bacterial community in raw milk for dairy cows. The widely used teat bath with iodine disinfectant was efficient in protecting teat skin but failed to restrict the potentially harmful dominant bacteria of *Pseudomonas*, which calls for further research on a more effective disinfectant.

**Supplementary Materials:** The following supporting information can be downloaded at https://www.mdpi.com/article/10.3390/fermentation8050235/s1: Figure S1. Core bacteria at the genus level in raw milk and on teat skin. (a) Heatmap of core bacteria at the genus level. Each row of the matrix represents one sample, while each column represents one bacterial taxon at the genus level. (b) Venn diagram of core bacteria at the genus level. A hypergeometric test was performed to illustrate the bacterial composition difference between raw milk and teat skin; Figure S2. Bacterial transfer contribution of core genera in raw milk and on teat skin. Each row of the matrix represents one sample, while each column represents one bacterial taxon at the genus level. The dominant bacteria at the genus level were developed from Figure 2; Figure S3. ANOSIM (analysis of similarities) of the bacterial community in raw milk and on teat skin. SkinA, teat skin microbiota before the pre-milking medicated teat bath; SkinB, teat skin microbiota after the pre-milking medicated teat bath; SkinC, teat skin microbiota after the post-milking medicated teat bath.

**Author Contributions:** Conceptualization, S.L. and S.J.; methodology, W.D. and C.G.; formal analysis, H.Y. and C.G.; data curation, S.J. and H.Y.; writing—original draft preparation, H.Y.; writing—review and editing, Y.Z., Y.W. and Z.C.; visualization, S.J. and H.Y.; supervision, S.L.; project administration, S.L. All authors read and agreed to the published version of the manuscript.

**Funding:** This research was funded by the National Dairy Industry and Technology System (CARS-36) and the National Natural Science Foundation of China (31772628).

**Institutional Review Board Statement:** The study was approved by the Ethical Committee of the College of Animal Science and Technology (Project number: 31772628) of China Agricultural University.

**Informed Consent Statement:** Not applicable.

**Data Availability Statement:** All sequences are available in the Genome Sequence Archive (https://bigd.big.ac.cn/gsa/ accessed on 1 March 2018) in the BIG Data Center under accession numbers CRA000780 and CRA000781, which were released at 2018-03-01.

**Acknowledgments:** The authors thank Fengning Farm and Hengshui Farm for their management of the animals and Yunzeng Zhang for language polishing.

**Conflicts of Interest:** The authors declare no conflict of interest.

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
