# Peer review of "Bacterial Composition and Interactions in Raw Milk and Teat Skin of Dairy Cows"

_fermentation, doi:10.3390/fermentation8050235_

Round 1

Reviewer 1 Report

In this manuscript, Yan, et al described the microorganism community from raw milk and teat skin, based on their high-throuhgput sequencing analysis. Studying the bacterial in raw milk is critical for food safety and agriculture. It is also interesting to study the source of bacteria in raw milk. First, the authors studied the community diversity, and compare them between raw milk and teat skin. The data also showed some special features of raw milk bacteria, such as higher centrality. Using Bayesian approach, the authors also studied the proportion of bacteria transferred between raw milk and teat skin. They finally discussed the dynamic shift during the milking procedures, which also help optimize the protocols.

All my concerns are minor, but still hope the authors can provide answers.

  1. sample collected from one location.

Samples from n=156 milk and n=313 teat skin were collected from two farms. But they seems to be from the same area, and probably close to each other. This may weaken the conclusion as the bacteria community may be significantly affected by the local environment, feeding materials. Do the authors have other independent samples? Collecting new samples may not be necessary. If the authors can find other literatures with samples collected from different regions, then they can compare in the discussion section.

2. Bacterial communities in teat skin

Three groups of samples were collected - before pre-milking teat, after pre-milking teat, after the post-milking tea bath. The authors combined them together, and studied the bacterial community distributions (Figure 1 and 2). Is there any difference among these groups? They all showed the same distributions? Combining them together may hide some specific features. This part may be trivial if the authors want to focus on the comparison between milk and teat skins.

Although the conclusion may be limited to certain area, this manuscript showed a strong bacterial community connection. Generally, the data were well presented with beautiful figures and the conclusions were strongly supported. The manuscript was well-prepared, and the topic is a good fit for the journal.

Author Response

  1. sample collected from one location.

Samples from n=156 milk and n=313 teat skin were collected from two farms. But they seems to be from the same area, and probably close to each other. This may weaken the conclusion as the bacteria community may be significantly affected by the local environment, feeding materials. Do the authors have other independent samples? Collecting new samples may not be necessary. If the authors can find other literatures with samples collected from different regions, then they can compare in the discussion section.

Response:

Thank you for your kind suggestions. we have added literature and comparations with samples collected from other regions in the discussion section, the changed description has been highlighted in red in the manuscript. we hope the current description could meet your criteria. This section has been modified as:

Understanding the movement of microbes from the environment into the raw milk is important for controlling the transfer of bacteria into the food chain. A previous study with samples from 3 cows and their surrounding environment, demonstrated that teat surface was the most significant source of contamination in raw milk, with herd feces being the next most prevalent source of contamination [8]; Vacheyrou, et al (2011) also speculated that the bacteria on the teat surface were the main sources of bacteria in raw milk for dairy cows by cultivation method[31]. These previous studies all demonstrated the teat skin as the main source of bacteria in raw milk, but as the limited samples and huge individual variations, knowledge of the bacterial transfer between raw milk and teat skin was still not quantitatively illustrated. We used 101 paired collected raw milk and teat skin swab samples before the pre-milking teat bath, observed that a majority of bacteria (92.1%±18.6%) in raw milk were transferred from teat skin, and about 63.6% of bacteria on teat skin might be transferred from raw milk. Although we did not sample the bacterial community in the surrounding environment, the high bidirectional transfer proportion of bacteria between milk and teat skin might also indicate the positive correlation between bacteria in teat skin and the teat cistern, and targeting the bacteria in teat skin might be the efficient method in controlling the transfer of bacteria into raw milk.

  1. Bacterial communities in teat skin

Three groups of samples were collected - before pre-milking teat, after pre-milking teat, after the post-milking tea bath. The authors combined them together, and studied the bacterial community distributions (Figure 1 and 2). Is there any difference among these groups? They all showed the same distributions? Combining them together may hide some specific features. This part may be trivial if the authors want to focus on the comparison between milk and teat skins.

Response:

Thank you for your kind suggestions. In Figure 1 and Figure 2, we only used 101 paired collected raw milk and teat skin swab samples before the pre-milking teat bath, and compared the bacteria in the pre-milking teat and raw milk. The reason to only use pre-milking teat skin is that bacteria in pre-milking teat represent the natural situations and is more important in playing the role to connect the surrounding environment and teat cistern, and previous studies also indicated the before pre-milk teat skin is the main source of bacteria in raw milk. We also compared the bacteria change in samples of the before pre-milking teat, after pre-milking teat, and after the post-milking tea bath in Figure 5 and Figure 6. In summary, we demonstrated the correlation between bacteria in the pre-milking teat and raw milk (Figure1-Figure4) and monitored the dynamic shift of the bacteria community on the skin during the milking procedure(Figure5-Figure6).

We are sorry for the confusion for you on the group assignment, we have added descriptions in method and result section, and the changed description has been highlighted in red in the manuscript.

Reviewer 2 Report

In this manuscript, the authors investigated the composition, diversity, and co-occurrence network of the bacterial communities in raw milk and on teat skin, as well as the shift of bacterial communities during the teat bath. The author suggested that the bacterial community on teat skin might be the target for manipulating the bacterial community in raw milk for dairy cows. The results are interesting, and some suggestions and comments are as follows:

  1. Figure 4b, indicate the meaning of red circle and blue circle;
  2. 3.5, The dynamic change of the bacterial taxa on the teat skin of cows before (Skin A) and after (Skin B) the pre-milking teat bath, as well as after the post-milking teat bath (Skin C) were monitored to assess the bacterial taxa shift, however, the bacterial taxa in the milk after the pre-milking and post-milking teat bath need to be checked. Is there any change in the milk after teat bath? Whether the milk quality was improved by teat bath?

Author Response

1. Figure 4b, indicate the meaning of red circle and blue circle;

Response:

Thank you for your kind suggestions. We have added descriptions in the legend of Figure 4b, “the red circle and blue circle represent bacteria in teat skin and raw milk respectively.”

2. 3.5, The dynamic change of the bacterial taxa on the teat skin of cows before (Skin A) and after (Skin B) the pre-milking teat bath, as well as after the post-milking teat bath (Skin C) were monitored to assess the bacterial taxa shift, however, the bacterial taxa in the milk after the pre-milking and post-milking teat bath need to be checked. Is there any change in the milk after teat bath? Whether the milk quality was improved by teat bath?

Response:

Thank you for your kind suggestions. In this work, the bacteria on teat skin is our major concern, so we aim to describe the bacterial profiles and bacterial interactions in raw milk and on teat skin, and monitor the dynamic changes of the bacterial taxa on teat skin during the teat bath process. This information has been highlighted in the instruction section. Additionally, in the standard milking procedure, cows are continually milked after a pre-milking teat bath, so the bacteria change information in the raw milk after the teat bath is hard to achieve, and the milk quality improvement assessment also needs another longitudinal study. Thank you for putting forward these issues, we would conduct some other studies on these concerns in the future.

Round 2

Reviewer 2 Report

The revised version is acceptable.